# Characteristics of Floral Volatiles and Their Effects on Attracting Pollinating Insects in Three *Bidens* Species with Sympatric Distribution

**DOI:** 10.3390/biology14101310

**Published:** 2025-09-23

**Authors:** Jun-Wei Ye, Jing-Lin Jia, Yong-Hong Xiao, Jia-Hui Zhou, Jian-Jun Zeng

**Affiliations:** 1Key Laboratory of Jiangxi Province for Biological Invasion and Biosecurity, School of Life Sciences, Jinggangshan University, Ji’an 343009, China; 2308301015@jgsu.edu.cn (J.-W.Y.); 2408301052@jgsu.edu.cn (J.-L.J.); zhoujiahui@jgsu.edu.cn (J.-H.Z.); 2School of Life Sciences, Jinggangshan University, Ji’an 343009, China; 920040049@jgsu.edu.cn

**Keywords:** *Bidens* species, plant-insect interactions, floral volatile organic compounds, pollinator-mediated selection, mating system evolution, reproductive assurance, β-Ocimene emission

## Abstract

**Simple Summary:**

A comparative analysis of floral scent composition and proportions among three invasive *Bidens* species with contrasting self-pollination abilities revealed that the outcrossing species, *Bidens pilosa* var. *radiata*, emits significantly higher levels of (E)- and (Z)-β-ocimene than its two selfing congeners. Y-tube olfactometer assays demonstrated that this diene mixture acts as a key signal attracting local generalist bees, ensuring ample visitation and maintaining a high outcrossing rate. These results indicate that during the evolutionary transition from outcrossing to selfing, plants actively reduce pollinator-”advertising” volatiles, thereby decreasing pollinator attraction and providing new insights into mating system evolution and plant-pollinator interactions.

**Abstract:**

The transition from outcrossing to self-pollination is an evolutionary process in angiosperms. However, the changes in floral volatile composition during this process and their impacts on the behavior of pollinators are poorly understood. Therefore, this study investigated the potential differences in the floral volatile profiles and pollinator attraction capabilities of three invasive *Bidens* species. The results indicated that *Bidens pilosa* var. *radiata* (BH), which serves as a transitional species between facultative outcrossing and obligate outcrossing attracts a greater diversity and abundance of pollinators such as *Apis cerana* compared to the more self-compatible *Bidens frondosa* (DL) and *Bidens pilosa* var. *pilosa* (SY). Furthermore, a total of 37, 33, and 34 Volatile Organic Compounds (VOCs) were identified in the floral volatiles of BH, DL, and SY, respectively, with no discernible trend of decreased number of floral VOCs owing to increased self-pollination ability. Moreover, eleven significantly different compounds in the floral volatiles of the three *Bidens* species were obtained. Among these (E)-β-Ocimene (18.31 ± 1.10%) and (Z)-β-Ocimene (33.93 ± 3.49%) in the floral volatiles of BH (52.24 ± 4.59%) was significantly higher than that of DL (1.72 ± 0.50%) and SY (0.32 ± 0.19%). Additionally, Y-tube olfactometer behavioral assays indicated that (E)- and (Z)-β-Ocimene significantly attracted *A*. *cerana.* These findings suggested that (E)- and (Z)-β-Ocimene contribute to the attractiveness of BH to local pollinators. Furthermore, it can be inferred that within *Bidens*, stronger self-pollination ability reduces the relative content of VOCs—such as (E)- and (Z)-β-Ocimene—used to attract generalist pollinators.

## 1. Introduction

Floral chemical signals comprising VOCs released by plants and they are rapidly recognized by pollinating insects [1,2,3]. Floral VOCs vary among populations and species [4,5] and pollinator preferences drive the diversification of floral VOCs [6,7,8,9,10]. The loss of self-incompatibility and the evolution of self-pollination are key strategies employed by plants to adapt with pollination [11,12]. Generally, selfing taxa have reduced quantity and lower emission levels of VOCs due to decreased reliance on pollinators [13]. For instance, compared to self-incompatible populations, the scent emission in self-pollinating populations of *Abronia umbellata* was reduced by 99% [14]. Oppositely, Majetic et al. [13]. discovered that self-compatible populations of *Phlox* released three VOCs and their total volatile emission rate was not significantly different from outcrossing populations. Likewise, Petrén et al. [15]. found that the evolution of self-compatibility does not necessarily select for reduced scent emission.

In new habitats, invasive plants can overcome various challenges such as reproduction which involves complex and diverse adaptation and evolutionary information [16,17,18]. self-compatibility provides a “reproductive assurance” advantage to invasive alien plants, which enables them to rapidly colonize even in the absence of pollinators through self-pollination [19]. However, many invasive alien plants exhibit mixed mating systems, and their reproductive success still depends on pollinators [20]. Do invasive species with mixed mating systems display characteristics such as reduced VOCs number, lower emission levels, and decreased reliance on pollinators as the degree of self-pollination increases? Invasive plant species can serve as an effective model to investigate the relationship between floral scent evolution and the loss of self-incompatibility mechanisms.

Invasive species such as *Bidens pilosa* var. *radiata*, *Bidens frondosa* and *Bidens pilosa* var. *pilosa* have caused significant ecological damage globally [21]. These species are widely distributed in southern China, and these species exhibit mixed mating systems. However, there are differences in their self-pollination abilities and outcrossing rates. For instance, BH shows a transition from facultative outcrossing to obligate outcrossing [22], while DL and SY have higher self-pollination abilities [23,24]. Their reproductive strategies for successful invasion make them valuable subjects for research into the relationship between floral scent evolution and the loss of self-incompatibility. This study hypothesized that invasive *Bidens* species with greater self-compatibility will: (i) exhibit a lower dependence on pollinating insects; (ii) exhibit a reduced number of floral VOCs; and (iii) possess a reduced content of compounds that attract generalist pollinators.

## 2. Materials and Methods

### 2.1. Sampled Species and Sites

The experiment was performed at Jinggangshan University in Ji’an, Jiangxi Province (27°07′01″ N, 115°01′34″ E, elevation 78 m) (Figure 1d). This site is situated in the middle reaches of the Ganjiang River and the central section of the Lohsiao Mountain. The experimental site has a subtropical monsoon humid climate with annual average temperature ranges from 17.1 to 18.6 °C, while the annual average precipitation is 1487 mm [25]. Three selected invasive species from the *Bidens* genus (*B. pilosa* var. *radiata*, *B. frondosa*, and *B. pilosa* var. *pilosa*) are dominant species within the community. Other prevalent wild flowering plants in the region include *Setaria viridis* L., *Sida rhombifolia* L., *Ipomoea triloba* L., and several additional species.

### 2.2. Pollinator Observations

The floral visitation behavior in three dominant populations of *Bidens* species during their peak flowering period (when more than 50% of the plants in the population were in bloom) was observed in September 2024. Meteorological conditions during the 6-day survey (12–18 September 2024) were: daily maximum temperature ranged 33.5–36.5 °C, relative humidity 64–78%, total precipitation 0.11 mm (15 September only), and mean wind speed 2.8–4.2 m·s^−1^. We selected a 2 m × 2 m quadrat for our observations and thirty flowers were randomly selected. The data were collected from 8:30 a.m. to 6:30 p.m. with a total of 10 h of observation every day over three consecutive days. The number of individual pollinators visiting the flowers, the duration of their stay, and the number of consecutive visits made by each pollinator were meticulously recorded. Additionally, insect nets were also employed to capture flower-visiting insects and prepare specimens. These specimens were subsequently brought back to the laboratory for species identification through a literature review and consultation with experts in the field of animal taxonomy [26]. Plant vouchers of the three *Bidens* taxa were deposited in the Herbarium of Jinggangshan University under accession number PJXJU202409, and insect vouchers were deposited in the Entomological Collection of Jinggangshan University under accession number IJXJU202409.

### 2.3. Sample Collections

The flowers free from dew, pesticide residues, and damage from pests or diseases were randomly selected on a sunny morning at 10:00 a.m. We collected 5 fully open flowers from each of the three species during their peak flowering period. Then, flowers were placed into 20 mL headspace vials, ensuring that the volume did not exceed half of the vial’s capacity and each plant species was replicated five times.

### 2.4. Identification of Floral Volatile Compounds

The SPME fiber (DVB/CAR/PDMS 50/30 µm, Supelco, Bellefonte, PA, USA) was conditioned at 250 °C for 30 min prior to use. It was then inserted into the headspace of each vial so that the fiber tip remained approximately 0.5 cm above the flowers without direct contact. The vials were incubated at 30 °C for 40 min to allow adsorption. After incubation, the fiber was withdrawn and immediately introduced into the gas chromatography-mass spectrometry (GC-MS) injection port (250 °C) for desorption and analysis.

Floral volatiles were analyzed and identified using a gas chromatograph-mass spectrometer (GC-MS; GC model QP2010, MS model QP2010; Shimadzu, Kyoto, Japan) with the following RTX-5 column (30 m × 320 μm × 0.25 μm). The injection was performed in splitless mode with a flow rate of 1 mL·min^−1^. The oven temperature was set as follows: the initial temperature was maintained at 50 °C for 3 min, then it increased to 100 °C at a rate of 2 °C·min^−1^, followed by an increase to 140 °C at a rate of 1 °C·min^−1^, and finally, it increased to 280 °C at a rate of 15 °C·min^−1^, for 5 min. The ion source was configured to scan mode with a scan range of 30 to 450 amu, utilizing electron ionization (EI) as the ionization method. The electron energy was set to 70 eV. The injection port was held at 250 °C, and high-purity helium served as the carrier gas. The ion source temperature was set to 290 °C, and the transfer line (interface) temperature to 250 °C.

The mass spectral data obtained from GC-MS analysis were utilized to search targeted compounds in the instrument’s built-in standard library (NIST17). With removing the peaks presented in control sample, the relative percentage content of each volatile component in the odor was calculated using the ion current peak area normalization method.

### 2.5. Olfactometer Behavioral Experiments

The transparent, colorless glass Y-tube features a base and two arms, each measuring 10 cm in length, with an angle of 60° between them and an inner diameter of 2.5 cm. The two arms of the Y-tube are sequentially connected to an odor source bottle, a filtered air device (which contains dry activated carbon), and a flowmeter. A filter paper strip (15 mm × 30 mm) saturated with 10 μL of the volatile solution was placed in one arm of the Y-tube, designated as the test arm, while the other arm containing 10 μL of liquid paraffin oil, served as the control arm [27].

A vacuum pump was utilized to regulate the airflow, with the flow rate set to 110 mL·min^−1^. For the behavioral assays, an *A. cerana* was introduced into the base of the Y-tube at a time and observed for 5 min. If a bee moved against the airflow along the base of the Y-tube and remained there for at least 2 min, it was classified as a responding bee. Bees that did not respond within the 5 min observation period were excluded from the analysis. Behavioral responses were measured between 9:00 a.m. and 5:00 p.m., at a temperature of 24 ± 2 °C and relative humidity of 70 ± 5%. Each dose of the differential volatile compounds was tested on 20 *A. cerana*, with each bee being used only once. After every 10 bees were tested, the inner and outer walls of the Y-tube were wiped with anhydrous ethanol, and dried, and the positions of the two arms of the Y-tube were swapped to eliminate any potential asymmetrical effects of the Y-tube.

### 2.6. Statistical Analysis

The relative content of each compound was calculated using Excel 2021 and results were expressed as the mean ± SE. The sampled site map was completed by ArcGIS 10.8 version. Principal component analysis (PCA) of the differential volatile compounds of the flowers of three *Bidens* species was conducted using the “vegan and ggplot2” package in R version 4.4.1. The Kruskal–Wallis nonparametric test was employed to assess the significance of differences among the principal components of the three species. PERMANOVA (Adonis, 999 permutations, Bray–Curtis distance) of the differential volatile compounds of the flowers of three *Bidens* species was conducted using the “vegan” package in R version 4.4.1. Comparative visualization analysis of the number of insect species visiting the three *Bidens* species was performed using the “UpSetR” package in R version 4.4.1. Using the built-in “stats” package in R version 4.4.1, we performed binomial GLMs to evaluate differences in the number of *Apis cerana* attracted to the odor source versus the control (CK) [28]. GraphPad Prism version 8.0.2 was utilized to visualize the relative content of different classes of floral volatiles across the three *Bidens* species. Origin 2021 was used to plot the visitation frequency of the primary flower-visiting insects associated with the three *Bidens* species. SIMCA version 14.1 was employed to perform partial least squares-discriminant analysis (PLS-DA) of the volatile components, with key volatile components selected based on variable importance in projection (VIP) values > 1.

## 3. Results and Analysis

### 3.1. Pollinator Composition and Visit Frequency

There were notable differences in the types of flower-visiting insects among the three *Bidens* species. BH showed the highest diversity of flower-visiting insects, comprising 9 species from 7 families within the orders of Hymenoptera, Diptera, and Lepidoptera. In contrast, DL has a relatively lower diversity, with 6 species from 3 families in Hymenoptera and Diptera. Furthermore, SY showed the least diversity, with only 1 species observed belonging to Hymenoptera order (Table 1). Regarding pollinator specificity, 6 pollinators were exclusive to BH, 3 pollinators were exclusive to DL, and *L. occidens* is the only pollinator that visits all three *Bidens* species. This suggests that different Bidens species possess distinct levels of attractiveness and selectivity for flower-visiting insects (Figure 2).

*A. cerana* and *L. resurgens* were the primary flower-visiting insects of BH, with relative abundances of 48% and 34%, respectively (Table 1). Their visitation frequency peaked at 13:30 p.m. with 0.60 ± 0.08 visits per hour per flower, and then it decreased gradually. Notably, a significant number of Hymenopteran flower-visiting insects were still observed at 17:30 p.m. at the field sites. In contrast to the other two *Bidens* species, only BH was found to attract pollinators from the order Lepidoptera. The pollination frequency of Lepidoptera peaked between 14:30 p.m. and 15:30 p.m., after which it declined, with no further sightings recorded after 16:30 p.m. (Table 1; Figure 3).

The primary flower-visiting insects in DL belonged to Dipteran species. This included *L. basalis*, *P. tricuspis*, and *E. arvorum*, with relative abundances of 26%, 33%, and 21%, respectively. The visitation frequency of Dipteran flower-visiting insects peaked between 11:30 a.m. and 12:30 p.m.; however, the overall frequency was low, averaging only 0.09 ± 0.08 visits per hour per flower. In contrast, only the Hymenopteran species *L. occidens* was observed visiting SY, with a visitation frequency peaking between 10:30 a.m. and 11:30 a.m. at 0.30 ± 0.05 visits per hour per flower (Table 1; Figure 3).

### 3.2. Composition and Categories of Floral Volatiles

A total of 37, 33, and 34 volatile compounds were identified in the flowers of BH, DL, and SY, respectively. The volatile components of BH flowers included 19 terpenes, 6 alkenes, 2 alkanes, 5 esters, 4 alcohols, and 1 ketone. The volatile components of DL flowers consisted of 16 terpenes, 6 alkenes, 3 alkanes, 1 ester, 6 aromatic compounds and 1 alcohol. The volatile components of SY flowers comprised 14 terpenes, 5 alkenes, 6 alkanes, 3 aromatic compounds, 2 alcohols and 4 ketones. Terpenes were the most abundant compounds in the floral volatiles of all three *Bidens* species, with relative contents of 82.45 ± 16.67%, 87.65 ± 5.35%, and 76.88 ± 38.02%, respectively (mean ± SD; Appendix A; Figure 4).

In the floral volatiles of BH, the compounds with relatively high relative contents included (Z)-β-Ocimene (33.93 ± 3.49%) and (E)-β-Ocimene (18.31 ± 1.10%). In the floral volatiles of DL, the compounds with relatively high relative contents included m-Cymene (52.23 ± 1.90%), β-Myrcene (11.97 ± 1.00%), and α-Phellandrene (11.17 ± 0.99%). Additionally, Sabinene (35.02 ± 6.79%) and (1R)-(+)-α-Pinene (27.19 ± 24.18%) were recognized as compounds with relatively high relative contents in SY floral volatiles (Appendix A).

### 3.3. Analysis of Differential Floral Volatiles

Principal component analysis (PCA) was performed on the floral odor compounds of the three *Bidens* species (Figure 5a). The floral odor compounds of BH, DL, and SY clustered distinctly, with no overlapping among the three species. The contribution rates of the first principal component (PC1) and the second principal component (PC2) were 29.2% and 25.3%, respectively, resulting in a cumulative contribution rate of 54.5%. These two principal components encapsulate the primary information regarding the floral volatiles of the three *Bidens* species. A Kruskal–Wallis nonparametric test was also conducted on the two principal components, revealing that the odors of the three *Bidens* species could be distinctly separated using the principal components PC1 and PC2 (χ^2^ = 7.2, *p* = 0.0273). These findings indicate significant differences in the floral volatile components among the three *Bidens* species (PERMANOVA: F_2_,_6_ = 44.70, R^2^ = 0.937, *p* = 0.004; Figure 5).

To identify the differential volatile compounds among the flowers of the three *Bidens* species, PLS − DA was utilized, employing variable importance in projection (VIP) value > 1 as the screening criterion. A total of 11 differential compounds were identified, including (E)-β-Ocimene, D-Limonene, α-Phellandrene, β-Myrcene, and α-Pinene (Table 2).

The PCA biplot results further demonstrate that C4 ((Z)-β-Ocimene), C5 ((E)-β-Ocimene), C6 (D-Limonene), C9 (α-Pinene), and C10 (Neo-alloocimene) are associated with BH; C1 (m-Cymene), C7 (α-Phellandrene), and C8 (β-Myrcene) are associated with DL; and C2 (Sabinene), C3 ((1R)-(+)-α-Pinene), and C11 (3-Carene) are associated with SY (Figure 5b). These differential compounds play a significant role in distinguishing the floral odor differences between the three plants, and they can serve as characteristic components for differentiating the floral odor profiles of the three Bidens species.

### 3.4. Behavioral Assays of Floral Volatiles in A. cerana

Among the 11 differential compounds identified, (Z)-β-Ocimene (10^−4^ g/mL) and (E)-β-Ocimene (10^−4^ g/mL) exhibited significant differences (*p* < 0.05) in their attraction effects on the primary pollinator (*A. cerana*) of BH compared to the control (CK, liquid paraffin) (*n* = 20; Table 3).

## 4. Discussion

The transition from outcrossing to selfing is regarded as one of the most prevalent evolutionary shifts in angiosperms [29]. Typically, selfing species exhibit similar reproductive traits in terms of morphology and function, collectively referred to as the “selfing syndrome.” This syndrome is characterized by a reduction in investment in traits associated with outcrossing, particularly those related to pollinator attraction, such as flower number and size [30,31,32]. In this study, the capitula of DL and SY consist of tubular flowers, while the capitula of BH comprise both tubular and ligulate flowers (Figure 1). The data indicate that BH significantly outperforms DL and SY in terms of pollinator attraction efficiency, as evidenced by an increase in the total visitation frequency of pollinating insects and the richness index of pollinator taxa. Notably, the locally generalist pollinator from the Hymenoptera order—*A. cerana*—was observed in large numbers in BH. Therefore, it can be inferred that the ligulate florets of BH play a crucial role in attracting pollinators, which is essential for its outcrossing reproductive success.

Existing research generally suggests that, compared to selfing species, outcrossing species are more inclined to increase their investment in traits related to pollinator attraction as part of their reproductive strategies, such as enhancing the diversity of floral volatile compounds. In contrast, selfing species or populations, due to their reproductive assurance mechanisms, typically exhibit a reduction in the diversity of floral volatile compounds [25,33,34,35]. This theory has been validated in some species, such as A. *umbellata* and *Arabidopsis thaliana* [14,36], where selfing populations have shown a significant decrease in the diversity of floral volatile compounds. However, in this study, DL and SY, had a higher propensity for selfing, and they did not show a significant reduction in the number of floral volatile compounds (33 and 34, respectively) compared to BH (37). Similarly, research by Majetic et al. [13]. on the floral volatiles of selfing species in the genus *Phlox* also supports the notion that floral volatiles undergo only minor changes rather than significant reductions. For instance, certain floral volatiles may possess larvicidal and insect-repelling properties. In fact, the functions of floral volatiles are not limited to attracting pollinators; they also play significant roles in plant defense and other ecological aspects [37]. For example, α-Muurolene has been confirmed to possess larvicidal and insect-repelling properties [38], while α-Phellandrene can attract the parasitic wasp *Aphidius ervi*, a natural enemy of pests [39]. Additionally, Sabinene exhibits significant antibacterial effects and has a repellent effect on *Triatoma rubrofasciata* [40,41]. In this study, the relative contents of α-Muurolene and Sabinene in the floral volatiles of SY were 3.28 ± 2.47% and 35.02 ± 6.79%, respectively. These volatile compounds may play significant roles in pest control. Furthermore, in the floral volatiles of DL, the relative content of α-Phellandrene reached 11.17 ± 0.99%, which is speculated to potentially attract parasitic wasps that serve as natural enemies of pests. These results indicate that the number of floral volatile compounds does not decrease with the enhancement of selfing ability, but it is likely to be influenced by multiple ecological functions.

Outcrossing species release a complex mixture of VOCs from their flowers, which are believed to enhance the visitation rates of pollinators. β-Ocimene is recognized as a compound that attracts generalist pollinators [42] and it significantly increases pollinator visitation rates [43]. Woźniak et al. [42] investigated the floral scent composition of selfing plants in the genus *Capsella* and found that the concentrations of (E)- and (Z)-β-Ocimene in the floral scent were significantly reduced, which aligns with the results of this study. Additionally, in this study, exhibited a higher rate of outcrossing, and had a combined relative content of (E)- and (Z)-β-Ocimene (52.24 ± 4.59%) in its floral volatiles. This was significantly higher than the combined relative contents of (E)- and (Z)-β-Ocimene in the floral volatiles of DL and SY, which have a greater capacity for autonomous selfing. It can be inferred that BH, as an invasive plant, (E)- and (Z)-β-Ocimene contribute to the attractiveness of BH to local pollinators such as *A*. *cerana*, thereby ensuring high seed set and outcrossing rates, which contribute to its reproductive success in the invaded area. This suggests the stronger selfing ability of a plant lowers the relative content of volatile compounds in its flowers used to attract pollinators.

## 5. Conclusions

This study reveals the impact of the evolutionary shift from outcrossing to selfing in plants on floral volatiles and pollination mechanisms. SY and DL, which exhibit a stronger tendency for selfing, did not show a significant reduction in the diversity of floral volatiles. This suggests that the composition of floral volatiles is not solely influenced by the ability to self-pollinate but may also be related to multiple ecological functions, such as plant defense.

BH had significantly higher relative contents of (E)- and (Z)-β-Ocimene in its floral volatiles compared to SY and DL. These volatile compounds can attract pollinators. As an invasive species, (E)- and (Z)-β-Ocimene enhance BH’s appeal to local pollinators, ensuring the success of outcrossing reproduction. This further confirms that plants with a tendency for selfing have lower relative content of volatile compounds in their flowers that attract pollinators. This study enriches our understanding of the evolution of selfing and outcrossing in plants and the functions of floral volatiles, providing a new perspective for comprehending the reproductive strategies and ecological adaptations of plants. However, we acknowledge the limitations of our study: although the relative abundance data reveal the dominant scent components, they do not permit quantitative comparison of emission rates among flowers or species; moreover, we lack comprehensive analysis of population dynamic history and the integration of methods such as Gas Chromatography-Electroantennographic Detection (GC-EAD). Such analyses could provide insights into the dynamic changes in floral volatile components at different flowering stages of sympatric distribution and their influence on pollinator attraction, as well as more accurately identify the characteristic compounds that attract different pollinators. Future research will focus on these issues, delving into how congeneric invasive plants in sympatric conditions achieve a balanced adaptive mechanism for niche preemption and reproductive assurance through the dynamic regulation of floral volatiles.

## Figures and Tables

**Figure 1 biology-14-01310-f001:**
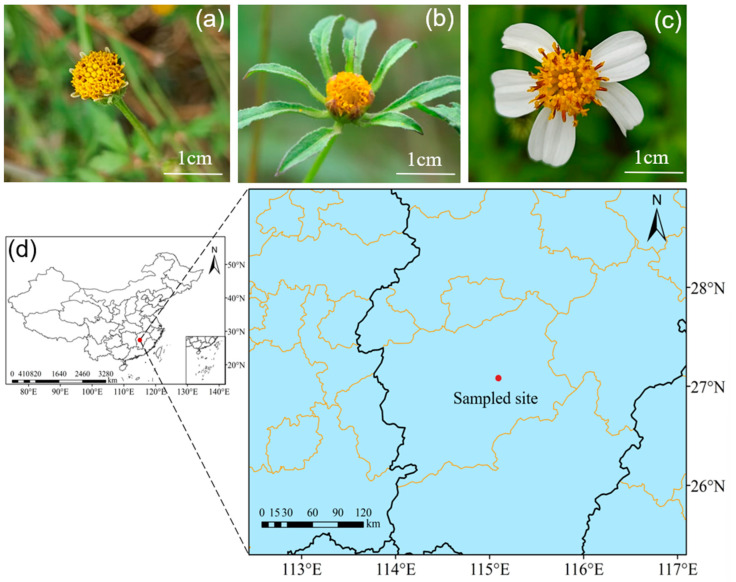
Three invasive *Bidens* species and Sampled site. (**a**). SY inflorescence with tubular flowers; (**b**). DL inflorescence with tubular flowers; (**c**). BH inflorescence with ligulate flowers and tubular flowers; (**d**). Sampled site.

**Figure 2 biology-14-01310-f002:**
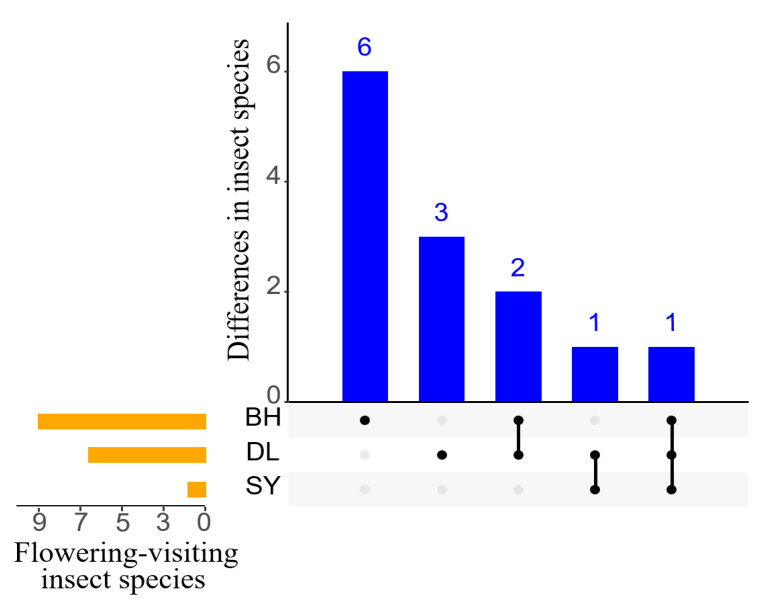
Comparative intersection diagram of flower-visiting insect species of three *Bidens*. Figure 2 is divided into three parts. The yellow bar chart illustrates the number of flower-visiting insect species associated with each plant species. The blue bar chart, along with the corresponding matrix scatter plot below it, displays the overlap of flower-visiting insect species among the three *Bidens* species.

**Figure 3 biology-14-01310-f003:**
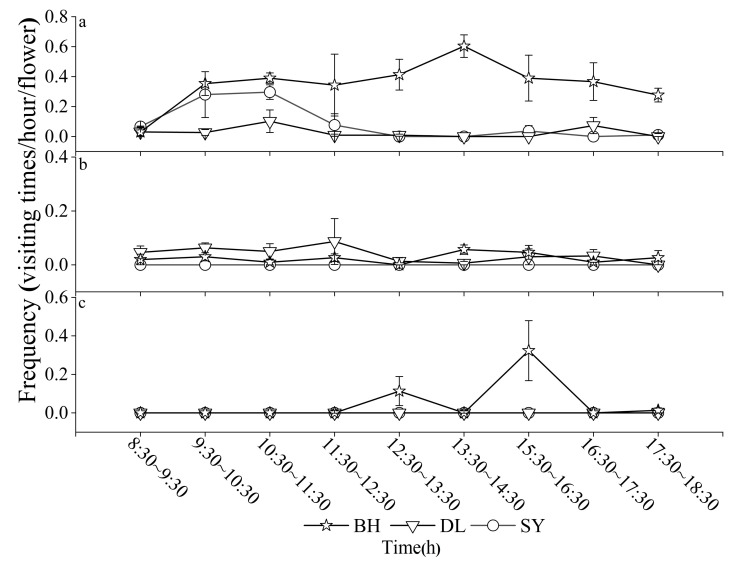
Visiting frequency of main flower-visiting insect groups of three *Bidens* (mean ± SE). (**a**) Hymenoptera; (**b**) Diptera; (**c**) Lepidoptera.

**Figure 4 biology-14-01310-f004:**
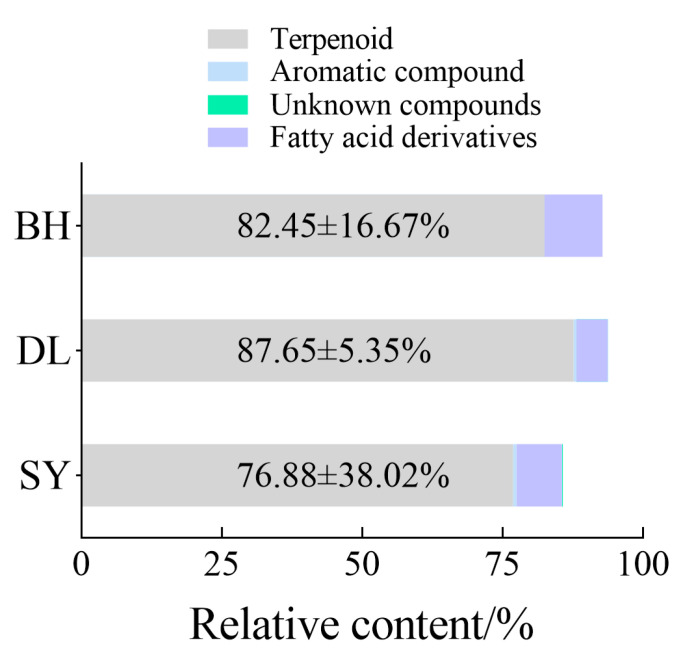
Relative content of different chemical classes of the volatile components from flower in three *Bidens* (*n* = 5).

**Figure 5 biology-14-01310-f005:**
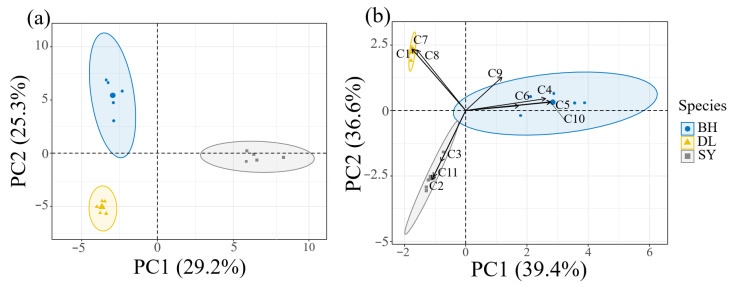
PCA correlation analysis of volatile compounds from flower in three *Bidens* (*n* = 5). (**a**): Volatile compounds; (**b**): Differential compounds.

**Table 1 biology-14-01310-t001:** Flower-visiting insects of three *Bidens*.

Order	Family	Genus	Species	Relative Abundance/%		
				BH	DL	SY
Hymenoptera	Halictidae	*Lasioglossum*	*L. occidens*	0.02	0.13	1.00
			*L. resurgens*	0.34	—	—
		*Megachile*	*Megachile* sp.1	0.01	0.02	—
	Apidae	*Apis*	*A. cerana*	0.48	—	—
Diptera	Phoridae	*Pseudacteon*	*P. tricuspis*	—	0.26	—
	Syrphidae	*Eristalinus*	*E. basalis*	—	0.33	—
			*E. arvorum*	0.05	0.21	—
			*E. baltea*	0.01	—	—
		*Helophilus*	*H. affinis*	—	0.05	—
	Papilionidae	*Graphium*	*G. sarpedon*	0.05	—	—
Lepidoptera	Lycaenidae	*Lampides*	*L. boeticus*	0.01	—	—
	Hesperiidae	*Pelopidas*	*P. sinensis*	0.02	—	—

Note: “—” indicates not observed.

**Table 2 biology-14-01310-t002:** Analysis of differential volatile compounds from flower in three *Bidens* (*n* = 5).

Code	VIP Value	Name	CAS	Average Relative Content/% (Mean ± SD)		
				BH	DL	SY
C1	3.97	m-Cymene	535-77-3	—	52.23 ± 1.90	—
C2	3.00	Sabinene	3387-41-5	—	—	35.02 ± 6.79
C3	2.55	(1R)-(+)-α-Pinene	7785-70-8	—	—	27.19 ± 24.18
C4	2.47	(Z)-β-Ocimene	13877-91-3	33.93 ± 3.49	1.49 ± 0.44	0.13 ± 0.05
C5	2.06	(E)-β-Ocimene	3779-61-1	18.31 ± 1.10	0.23 ± 0.06	0.19 ± 0.14
C6	2.04	D-Limonene	5989-27-5	5.87 ± 4.45	—	—
C7	1.82	α-Phellandrene	99-83-2	—	11.17 ± 0.99	—
C8	1.61	β-Myrcene	123-35-3	3.25 ± 3.22	11.97 ± 1.00	2.54 ± 0.62
C9	1.53	α-Pinene	80-56-8	8.67 ± 1.47	8.41 ± 0.41	—
C10	1.48	Neo-alloocimene	7216-56-0	9.91 ± 0.58	0.09 ± 0.04	—
C11	1.09	3-Carene	13466-78-9	—	—	3.59 ± 1.35

Note: “—” indicates not detected.

**Table 3 biology-14-01310-t003:** Detection of 11 differential floral volatile compounds and their attraction differences to *A*. *cerana*: Y-tube olfactometer bioassay analyzed with Binomial GLMs.

Component	Dosage (g/mL)	Number of Bees to Choose Odors	Number of Bees to Choose Paraffin	Binomial GLMs
(Z)-β-Ocimene	10^−2^	12	8	β = 0.41 ± 0.45, *p* = 0.374
	10^−4^	15 *	5	β = 1.10 ± 0.50, *p* = 0.029 *
	10^−6^	12	8	β = 0.41 ± 0.4, *p* = 0.374
(E)-β-Ocimene	10^−2^	12	8	β = 0.41 ± 0.45, *p* = 0.374
	10^−4^	16 *	4	β = 1.39 ± 0.56, *p* = 0.013 *
	10^−6^	8	12	β = –0.41 ± 0.45, *p* = 0.374
D-Limonene	10^−2^	13	7	β = 0.62 ± 0.47, *p* = 0.190
	10^−4^	11	9	β = 0.20 ± 0.45, *p* = 0.655
	10^−6^	12	8	β = 0.41 ± 0.45, *p* = 0.374
α-Phellandrene	10^−2^	10	10	β = 0.00 ± 0.45, *p* = 1.000
	10^−4^	14	6	β = 0.85 ± 0.48, *p* = 0.074
	10^−6^	6	14	β = –0.85 ± 0.48, *p* = 0.074
Sabinene	10^−2^	11	9	β = 0.20 ± 0.45, *p* = 0.655
	10^−4^	10	10	β = 0.00 ± 0.45, *p* = 1.000
	10^−6^	9	11	β = –0.20 ± 0.45, *p* = 0.655
3-Carene	10^−2^	13	7	β = 0.62 ± 0.47, *p* = 0.190
	10^−4^	13	7	β = 0.62 ± 0.47, *p* = 0.190
	10^−6^	10	10	β = 0.00 ± 0.45, *p* = 1.000
α-Pinene	10^−2^	10	10	β = 0.00 ± 0.45, *p* = 1.000
	10^−4^	9	11	β = –0.20 ± 0.45, *p* = 0.655
	10^−6^	7	13	β = –0.62 ± 0.47, *p* = 0.190
(1R)-(+)-α-Pinene	10^−2^	14	6	β = 0.85 ± 0.48, *p* = 0.074
	10^−4^	12	8	β = 0.41 ± 0.45, *p* = 0.374
	10^−6^	13	7	β = 0.62 ± 0.47, *p* = 0.190
m-Cymene	10^−2^	11	9	β = 0.20 ± 0.45, *p* = 0.655
	10^−4^	8	12	β = –0.41 ± 0.45, *p* = 0.374
	10^−6^	9	11	β = –0.20 ± 0.45, *p* = 0.655
β-Myrcene	10^−2^	9	11	β = –0.20 ± 0.45, *p* = 0.655
	10^−4^	9	11	β = –0.20 ± 0.45, *p* = 0.655
	10^−6^	8	12	β = –0.41 ± 0.45, *p* = 0.374
Neo-alloocimene	10^−2^	11	9	β = 0.20 ± 0.45, *p* = 0.655
	10^−4^	8	12	β = –0.41 ± 0.45, *p* = 0.374
	10^−6^	11	9	β = 0.20 ± 0.45, *p* = 0.655

* significance.

## Data Availability

The datasets generated for this study are available on request to the corresponding author.

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
