# Peer review of "Characteristics of Floral Volatiles and Their Effects on Attracting Pollinating Insects in Three Bidens Species with Sympatric Distribution"

_biology, 2025, doi:10.3390/biology14101310_

Round 1

Reviewer 1 Report

Comments and Suggestions for Authors

The authors compare pollinator communities and floral volatile organic compounds (VOCs) among three invasive Bidens species. The topic is both timely and interesting, as it links insect biology with floral scent chemistry. However, several aspects need substantial clarification or revision before the paper can be accepted:

Please add retention index on the same stationary phase (report calculated RIs, reference RIs, and ΔRI thresholds) and indicate where identities were confirmed with authentic standards (especially for the markers: β-ocimenes, m-cymene, sabinene, (1R)-(+)-α-pinene, D-limonene, α-phellandrene, β-myrcene, α-pinene, neo-alloocimene, 3-carene

Peak-area normalization alone cannot compare emission amounts. Please report absolute or semi-quantitative emission rates per flower (ng h⁻¹ flower⁻¹) or, even better, on floral mass.

Specify SPME fiber coating, conditioning protocol, sampling order, blank controls and report the number of flowers per vial.

The methods state “GC model QP2010, MS model HP-5977; Shimadzu, Japan.” The QP2010 is typically a GC–MS (Shimadzu), and HP-5977 is an Agilent MS. This combination is unusual—please verify and correct the instrument model(s).

Please provide the MS source temperature and transfer line temperature (you list “MS temperature 150 °C”; I believe this is the interface).

Please confirm column ID is 0.32 mm.

For Y-tube choices, a binomial GLM or exact binomial test is preferable to χ² against a 0.5 expectation.

Provide voucher deposition for insects (institution + accession numbers) and plants (herbarium vouchers for the three Bidens taxa)

Please verify all Latin names, authorities, and current taxonomy and italicize genera/species throughout.

The abstract and discussion state both that there is no trend of reduced VOC complexity with increased selfing and that stronger selfing decreases the abundance of volatile compounds used to attract generalist pollinators. The latter assertion is not supported by relative fragrance data alone.

Phrases like “BH relies on (E)- and (Z)-β-ocimene” overstate what lab Y-tube tests can show. You haven’t done GC-EAD or field manipulations.

You equate “number of compounds” with complexity. Diversity metrics (richness, Shannon) weren’t analyzed.

For all figures, provide measures of dispersion (mean ± SE, already used) with per-replicate points for VOC classes and key compounds.

Comments on the Quality of English Language

English needs careful editing throughout (hyphenation, spacing, tense consistency).

Reviewer 2 Report

Comments and Suggestions for Authors

Dear Authors

Please check my comments in pdf file. 

Round 2

Reviewer 1 Report

Comments and Suggestions for Authors

 Provided the concerns are resolved in a revised version, the paper should be accepted.

Author Response

Dear Reviewer,

  Thank you very much for your positive feedback and for stating that the paper "should be accepted" once the listed concerns are resolved. We appreciate your confidence in our work.

  In this revised version, we have carefully addressed every point you raised. All changes are described in detail in the point-by-point response letter and are highlighted in red in the revised manuscript. We believe that the revised paper now fully meets the requirements for publication.

  Thank you again for your time and constructive comments.

Yours sincerely, 

Jun-Wei Ye
  On behalf of all co-authors.

Reviewer 2 Report

Comments and Suggestions for Authors

Dear Authors

Please add  temperature, humidity, precipitation, wind) recorded during the observation periods in material and method section.  

Thanks

Author Response

Sorry,We have made revisions on the basis of the original version; please take this edition as the final one.

(Q1) Please add temperature, humidity, precipitation, wind) recorded during the observation periods in material and method section.

Dear Reviewer, 

Thank you for your questions. We have now inserted a concise sentence in the materials and methods section (lines99–102) “Meteorological conditions during the 6-day survey (12–18 September 2024) were: daily maximum temperature ranged 33.5–36.5 °C, relative humidity 64–78%, total precipitation 0.11 mm (15 Sep only), and mean wind speed 2.8–4.2 m·s⁻¹.

The full daily data set is provided in the attachment. All relevant revisions have been highlighted in red in the revised manuscript.
